# Dissolved nitrogen uptake versus nitrogen fixation: Mode of nitrogen acquisition affects stable isotope signatures of a diazotrophic cyanobacterium and its grazer

**Michelle Helmer**[1,2], **Desiree Helmer**[1], **Elizabeth Yohannes**[1,3], **Jason Newton**[4], **Daniel R. Dietrich**[5], **Dominik Martin-Creuzburg**[6]*

1 University of Konstanz, Limnological Institute, Konstanz, Germany, 2 Wahnbachtalsperrenverband, Siegburg, Germany, 3 Department of Bird Migration, Swiss Ornithological Institute, Sempach, Switzerland, 4 National Environmental Isotope Facility, Scottish Universities Environmental Research Centre, East Kilbride, United Kingdom, 5 University of Konstanz, Human and Environmental Toxicology, Konstanz, Germany, 6 Department of Aquatic Ecology, BTU Cottbus-Senftenberg, Research Station Bad Saarow, Bad Saarow, Germany

* Dominik.Martin-Creuzburg@b-tu.de

**Data Availability Statement:** The raw data supporting the results presented in this study are publicly available in the Zenodo repository at

## Abstract

Field studies suggest that changes in the stable isotope ratios of phytoplankton communities can be used to track changes in the utilization of different nitrogen sources, i.e., to detect shifts from dissolved inorganic nitrogen (DIN) uptake to atmospheric nitrogen ($N_2$) fixation by diazotrophic cyanobacteria as an indication of nitrogen limitation. We explored changes in the stable isotope signature of the diazotrophic cyanobacterium *Trichormus variabilis* in response to increasing nitrate ($NO_3^-$) concentrations (0 to 170 mg L$^{-1}$) under controlled laboratory conditions. In addition, we explored the influence of nitrogen utilization at the primary producer level on trophic fractionation by studying potential changes in isotope ratios in the freshwater model *Daphnia magna* feeding on the differently grown cyanobacteria. We show that $\delta^{15}N$ values of the cyanobacterium increase asymptotically with DIN availability, from -0.7 ‰ in the absence of DIN (suggesting $N_2$ fixation) to 2.9 ‰ at the highest DIN concentration (exclusive DIN uptake). In contrast, $\delta^{13}C$ values of the cyanobacterium did not show a clear relationship with DIN availability. The stable isotope ratios of the consumer reflected those of the differently grown cyanobacteria but also revealed significant trophic fractionation in response to nitrogen utilization at the primary producer level. Nitrogen isotope turnover rates of *Daphnia* were highest in the absence of DIN as a consequence of $N_2$ fixation and resulting depletion in $^{15}N$ at the primary producer level. Our results highlight the potential of stable isotopes to assess nitrogen limitation and to explore diazotrophy in aquatic food webs.

## Introduction

Phosphorus (P) is commonly considered the most important nutrient limiting phytoplankton growth in freshwater ecosystems [1–3], despite ample evidence that freshwater phytoplankton

https://zenodo.org/records/11501652 (DOI: 10.5281/zenodo.11501651).

**Funding:** The work was financially supported by the Ministry of Science, Research, and the Arts of the Federal State Baden-Württemberg, Germany (Water Research Network Project: Challenges of Reservoir Management—Meeting Environmental and Social Requirements, to D.M.-C. and D.D.) and by the Deutsche Forschungsgemeinschaft (DFG, German Research Foundation, 298726046/GRK2272, to D.M.-C. and E.Y.). The authors declare no conflict of interest. The funders had no role in the design of the study; in the collection, analyses, or interpretation of data; in the writing of the manuscript; or in the decision to publish the results.

**Competing interests:** The authors have declared that no competing interests exist.

are just as frequently nitrogen (N)- as P-limited [4, 5]. In strongly N-limited water bodies, the phytoplankton community is typically dominated by diazotrophic bloom-forming cyanobacteria [6, 7]. Diazotrophic cyanobacteria can overcome N limitation by fixing atmospheric nitrogen ($N_2$). $N_2$ fixation is an oxygen-sensitive process and therefore either temporally or spatially separated from oxygenic photosynthesis. In filamentous cyanobacteria, $N_2$ is often fixed in highly specialized cells, the heterocysts [8]. $N_2$ fixation is energetically more expensive than the assimilation of dissolved inorganic N (DIN) from nitrate ($NO_3^-$) or ammonium ($NH_4^+$), potentially resulting in decreased growth [9–12]. Heterocyst differentiation is thus supposed to increase with decreasing DIN concentrations [13].

Stable isotopes have been extensively used to explore carbon (C) and nitrogen fluxes as well as trophic relationships in aquatic food webs [14, 15]. $^{15}N$ has been proposed to be an excellent tracer for fixed atmospheric N in ecosystems because $N_2$ is often depleted in $^{15}N$ compared to nitrate ($NO_3^-$) and other dissolved N sources [16, 17], and because diazotrophic cyanobacteria discriminate against $^{15}N$ during $N_2$ fixation [18, 19]. Atmospheric $N_2$ is used as a reference gas for N isotope measurements, and the $\delta^{15}N$ value of $N_2$ is therefore 0 ‰. The discrimination against $^{15}N$ during $N_2$ fixation typically leads to slightly negative $\delta^{15}N$ values (-1 to -2 ‰) in diazotrophic cyanobacteria [20]. The $\delta^{15}N$ values of $NO_3^-$ in aquatic systems typically range between +7 and +20 ‰, depending on N sources and biological transformations [21]. Negative $\delta^{15}N$ values of $NO_3^-$ may result from atmospheric $NO_3^-$ deposition or nitrification in soils, reflecting the complexity of isotopic transformations within the nitrogen cycle [22]. Nonetheless, particulate organic matter produced with the aid of N fixation typically acquires a distinctive isotope signature that reflects the isotope signature of $N_2$ and the discrimination against $^{15}N$ during N fixation, and this signature is traceable in the food web [23–28]. Consequently, the N isotope signature of natural seston can reveal $N_2$ fixation by cyanobacteria [23], while the zooplankton isotope signature can reflect the assimilation of organic matter produced by N-limited, diazotrophic cyanobacteria [24, 26, 28, 29]. This information has been used to estimate diazotrophic inputs to net N assimilation at different trophic levels of the food web [23, 25, 30]. Both N and P limitation can also influence the C isotope signature of photosynthetic organisms through changes in the activity of the $CO_2$ concentrating mechanism [31–35], but it remains unclear how the mode of N acquisition affects C stable isotope values of diazotrophs and their consumers.

The impact of N acquisition on the N and C isotope signature of primary producers and the traceability of this isotope change at the consumer level has rarely been studied experimentally. We cultivated the diazotrophic cyanobacterium *Trichormus variabilis* (formerly known as *Anabaena variabilis*) under controlled laboratory conditions at different dissolved nitrate ($NO_3^-$) concentrations, ranging from 0 to 170 mg $L^{-1}$, and analyzed changes in N and C stable isotope values ($\delta^{15}N$ and $\delta^{13}C$) in cyanobacterial cells. In addition, to assess the trophic transfer of the experimentally generated differences in isotope signatures, we analyzed the N and C stable isotope values of the freshwater model herbivore *Daphnia* after feeding on the differently grown cyanobacteria. We hypothesized that a reduction in DIN availability is associated with a switch to $N_2$ fixation in *T. variabilis*, and that this change in N acquisition is reflected both in $\delta^{15}N$ and $\delta^{13}C$ values of *T. variabilis* and *Daphnia*. Our data show that the availability of DIN strongly affects the N isotope values of the diazotrophic cyanobacterium *T. variabilis*, pointing towards increasing $N_2$ fixation with decreasing dissolved N availability. We also show that these differences in isotope signatures are detectable at the consumer level despite varying trophic fractionation, highlighting the potential of stable isotopes to explore the significance of diazotrophy in aquatic food webs.

## Materials and methods

### Experimental setup

To study the differential effects of N fixation and $NO_3^-$ assimilation on stable isotope signatures of a diazotrophic cyanobacterium experimentally, we cultured *Trichormus variabilis* P9 (ATCC 29413) at four different dissolved N ($NaNO_3$) concentrations (Table 1). *T. variabilis* was cultured semicontinuously in 1-L flasks containing modified Woods Hole Medium without vitamins [36] with ¼ of the medium being replaced daily. The flasks (n = 4) were exposed to a light: dark cycle (16 h: 8 h) with illumination at 180 µmol quanta $m^{-2} s^{-1}$ at 20 ˚C. The cultures were continuously aerated with sterile-filtered ambient air (source of $N_2$). Cyanobacterial cells were harvested after nine weeks of growth during their exponential growth phase (estimated from optical density measurements) and filtered onto pre-combusted GF/F filters (Whatman™, GE Healthcare Life Science, Chicago, USA). Filters were dried at 50 ˚C and stored in a desiccator until subsequent stable isotope measurement.

Stock cultures of a clone of *Daphnia magna* [37] were maintained on filtered lake water (0.2 µm pore-sized membrane filter) and saturating concentrations of the green alga *Acutodesmus obliquus* (SAG 276-3a), which was cultured in Cyano Medium [38] in 5-L batch cultures under permanent illumination. The growth experiment was conducted at 20 ˚C with a cohort of third-clutch neonates born within 12 h which were reared on saturating amounts (2 mg C $L^{-1}$) of *A. obliquus* in glass beakers containing 200 ml of filtered lake water (0.2 µm pore-sized membrane filter). After five days of feeding on *A. obliquus*, the animals were transferred to beakers containing 2 mg C $L^{-1}$ of the cyanobacterium *T. variabilis*, which was cultivated at the different DIN concentrations. Food suspensions were prepared from the different DIN treatments through centrifugation and resuspension of the cells in an aliquot of filtered lake water. Carbon concentrations of the different food suspensions were estimated from photometric light extinction (480 nm) and from carbon extinction equations determined prior to the experiment. Each treatment (i.e., nitrogen concentration) initially consisted of 30 beakers with five *D. magna* in each beaker. The experimental animals were transferred daily into new beakers containing freshly prepared food suspensions. Before the diet switch ($T_0$) at day five and after the diet switch daily over a period of four days, six beakers of each treatment were randomly subsampled (5 samplings × 6 beakers = 30 beakers). The sampled animals were stored frozen in reaction tubes. After the experiment all samples were dried for 24 h, weighed on an electronic balance (Sartorius 4504MP8; ± 0.1 µg) and stored in a desiccator until subsequent stable isotope analysis.

### Stable isotope analysis

The dried *Daphnia* samples (0.3–0.7 mg) were transferred into pre-weighed tin cups and weighed to the nearest of 0.0001 mg using a micro-balance (Sartorius 4504MP8). The dried GF/F filters loaded with cyanobacteria were also transferred into tin cups for subsequent stable

**Table 1. Sodium nitrate ($NaNO_3^-$) and resulting dissolved inorganic nitrogen (DIN) concentrations as well as molar N:P ratios of the growth medium used to cultivate the diazotrophic cyanobacterium Trichormus variabilis.** The phosphorus concentration of the medium was 2.03 mg $L^{-1}$.

| $NaNO_3^-$ [mg $L^{-1}$] | DIN [mg $L^{-1}$] | $N:P_{molar}$ |
|---|---|---|
| 0 | 0 | 0 |
| 42.5 | 7 | 8 |
| 85 | 14 | 15 |
| 170 | 28 | 31 |

isotope analysis. Encapsulated samples were combusted using a Elementar vario PYRO cube® elemental analyzer (2013), and the resulting purified $N_2$ and $CO_2$ gases analyzed for $\delta^{15}N$ and $\delta^{13}C$ respectively on a Thermo Fisher Delta XP Plus Isotope Ratio Mass Spectrometer. Ratios were corrected for instrument drift and linearity using regularly interspersed internal laboratory standards (gelatin, glycine and alanine mixtures) with known stable isotope values (for details see [39]. Stable isotope ratios were expressed relative to Vienna Pee Dee Belemnite ($\delta^{13}C$) or air ($\delta^{15}N$). Precision of the measurements was 0.09 ‰ for $\delta^{13}C$ and 0.15 ‰ for $\delta^{15}N$, based on the standard deviation of the most common laboratory standard used (gelatin) over the five-isotope analysis runs.

Stable isotopes data are reported ($\delta^{13}C$ and $\delta^{15}N$) in parts per thousand (‰) using the $\delta$-notation of McKinney et al. [40]:

$$\delta X(‰) = 1000 \times \left[ \left( \frac{R_{sample}}{R_{standard}} \right) - 1 \right]$$

X is $^{13}C$ or $^{15}N$,
$R_{sample}$ is the $^{13}C/^{12}C$ and $^{15}N/^{14}N$ ratios of our samples and
$R_{standard}$ is that of international standards V- PDB, AIR, respectively.

A total of 60 *Daphnia* samples (3 replicates of each treatment and time point) and 16 phytoplankton samples (a single replicate of each time point) were subjected to stable isotope analysis.

**Turnover rates.** The turnover rates were calculated from the changes in $\delta^{13}C$ and $\delta^{15}N$ values after the diet switch using the exponential model of Hobson and Clark [41]:

$$\delta_t = \delta_{eq} + \left( \delta_0 - \delta_{eq} \right) e^{-\lambda t}$$

$\delta_t$ is the $\delta^{13}C$ and the $\delta^{15}N$ value of the daphnids at experimental time t,
$\delta_{eq}$ is the calculated asymptotic equilibrium with the new diet,
$\delta_0$ is the initial isotope value prior to the diet-switch, and
$\lambda$ is the turnover rate ($h^{-1}$).

For the estimation of the variables in this exponential model we used the nls function in R with the self-starting Asymptotic Regression Model SSasymp based on the equation:

$$\delta_t = \delta_{eq} + \left( \delta_0 - \delta_{eq} \right) e^{-\exp(\log \lambda)t}$$

We expressed the turnover rates as the time period needed to achieve a 50% turnover of isotope composition of $\delta^{13}C$ and $\delta^{15}N$ (half-life, $T_{0.5}$) using the function of Hobson and Clark [41]:

$$T_{50} = \frac{\ln(2)}{\lambda}$$

**Time depending isotope signature analysis of *Daphnia*.** To analyze the adaptation of *Daphnia* to its dietary source over time, we used the equation:

$$\delta_{(t4-t0)} = \delta_{96h\ after\ diet\ switch} - \delta_{initial}$$

**Trophic fractionation.** The trophic fractionation was determined by subtracting dietary isotope values from those of the consumer:

$$\Delta_{(cons-diet)} = \delta_{cons} - \delta_{diet}$$

## Statistical analysis

All statistical analyses were performed using R (version 3.6.2). The data were checked for normality (Shapiro-Wilk Test) followed by the test for homogeneity of variances (Levene's Test). Treatment effects (n = 4) were analyzed using ANOVA followed by Tukey's HSD post hoc test. Pearson's method was applied to investigate correlations between C and N content (%) of the animals and their stable isotope signature. Non-linear correlation was performed using the R package "nlcor" [42]. For analyzing the dose response curve, we used the asymptotic regression model of the R package "drc" [43] with the function AR.3(). The model is a three-parameter model with the function:

$$f(x) = c + (d - c)(1 - e^{\frac{-x}{e}})$$

c is the lower limit
d is the upper limit, and
e > 0 is determining the steepness of the increase as x.
The median effective dose ($ED_{50}$) was estimated using the ED function of the model.

## Results

### Nitrate availability and stable isotope values of *T. variabilis*

The $\delta^{15}N$ values of *T. variabilis* were positively correlated with the DIN concentrations in the growth medium (Fig 1a). The lowest $\delta^{15}N$ was found in the absence of DIN (-0.7 ‰). The asymptotic regression model revealed a maximum $\delta^{15}N$ of 3.0 ‰ (± 0.2; $p < 0.001$), a minimum $\delta^{15}N$ of -0.7 ‰ (± 0.2; $p < 0.01$) and a median effective dose ($ED_{50}$) of 3.6 mg $L^{-1}$ (± 0.7) of DIN. The residual standard error of this model fit amounts to 0.38. While the $\delta^{15}N$ values of *T. variabilis* were positively correlated with DIN, the $\delta^{13}C$ values did not show a clear relationship with DIN availability (Fig 1b). The $\delta^{13}C$ values of *T. variabilis* grown at a

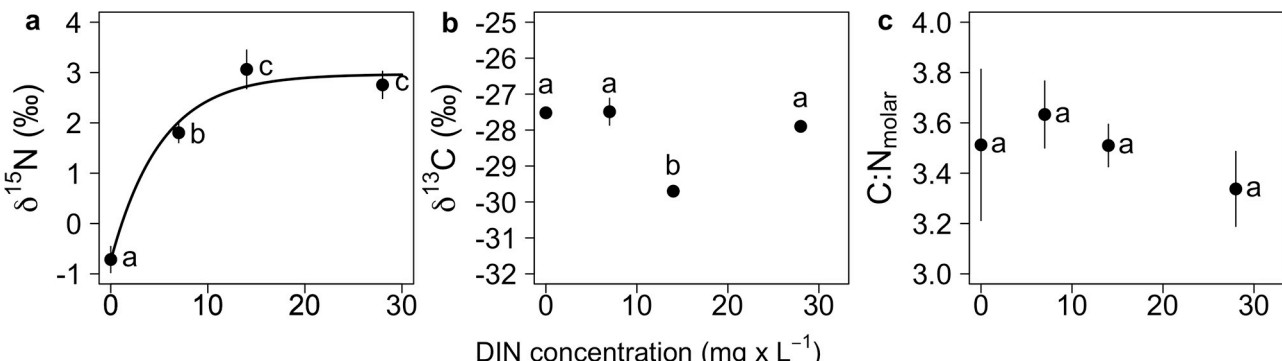

**Fig 1. a** Asymptotic regression model showing the relationship between dissolved inorganic nitrogen (DIN) concentrations in the growth medium and $\delta^{15}N$ values of *T. variabilis*. **b** Relationship between DIN concentrations and $\delta^{13}C$ of *T. variabilis*. **c** Relationship between DIN concentrations and particulate molar C:N ratios of *T. variabilis*. Letters show statistically significant differences between data points (Tukey's HSD following ANOVA).

DIN concentration of 14 mg L$^{-1}$ were significantly lower (around 2 ‰) than the $\delta^{13}$C values of *T. variabilis* grown at the other DIN concentrations (Tukey's HSD: p < 0.001; ANOVA: $F_{3,11}$ = 110.7, p < 0.001). Carbon concentrations in the different DIN treatments (mean 0.19 ± 0.02 mg ml$^{-1}$) did not differ significantly (Kruskal-Wallis Test: $\chi^2$ = 6.49, df = 3, p > 0.05). Molar C:N ratios of *T. variabilis* were consistently low (mean 3.49 ± 0.2) and did not change significantly with DIN concentrations (ANOVA, $F_{3,11}$ = 0.07, p > 0.05; Fig 1c).

### Stable isotope values of *Daphnia* in relation to DIN availability

The mode of N acquisition at the primary producer level (*T. variabilis*) also influenced the isotope signature of *Daphnia* consuming the cyanobacteria that were grown at the different DIN concentrations (Fig 2). The stable isotope values of *Daphnia* were significantly positively correlated with those of their diet (N: r = 0.96, p < 0.001; C: r = 0.77, p = 0.002; Fig 2a and 2b).

The $\delta^{15}$N values of *Daphnia* increased with DIN concentration from 5.4 ‰ at 0 mg L$^{-1}$ DIN to 8.2 ‰ at 28 mg L$^{-1}$ DIN after the diet switch (Fig 3a). The isotope N signature of the animals (96 h after diet switch) correlated positively with the N content of the diet (Pearson: r = 0.69, p = 0.01). We did not find any correlation of isotope N signature of *Daphnia* and molar C:N ratio of the diet. The $\delta^{13}$C values of *Daphnia* did not show a clear relationship with

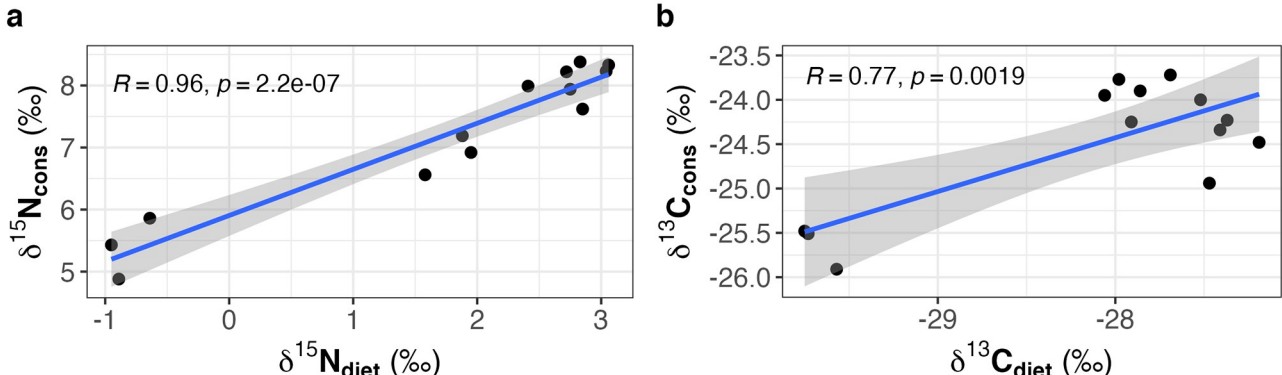

**Fig 2.** Stable isotope values (**a:** $\delta^{15}$N and **b:** $\delta^{13}$C) of the consumer *Daphnia* in relation to those of their cyanobacterial diet cultivated at different DIN concentrations.

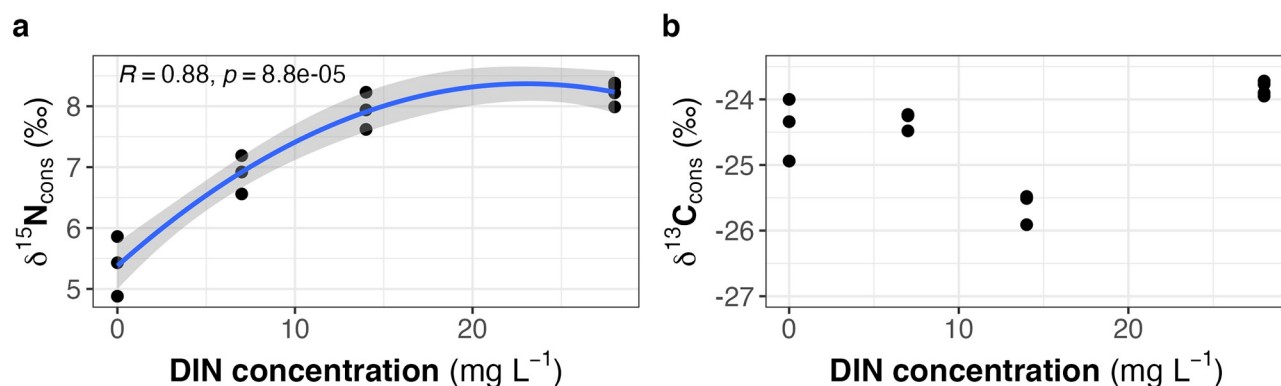

**Fig 3.** Stable isotope values (**a:** $\delta^{15}$N and **b:** $\delta^{13}$C) of *Daphnia* in relation to the different DIN concentrations in the cyanobacterial culture medium.

DIN concentration in the cyanobacterial culture medium; the obtained cosine function reflected the dietary signature (Figs 1b and 3b). We did not find any correlation between the isotope C signature of *Daphnia* and the elemental composition of the diet (N and C content or molar C:N ratios).

## Temporal changes in stable isotope values of *Daphnia*

$\delta^{15}$N values of *Daphnia* decreased over time while feeding on the cyanobacteria, most evident at DIN concentrations $\leq 7$ mg L$^{-1}$ (Fig 4a). The N isotope turnover rates increased (decreasing $T_{50}$ values) with increasing DIN concentration in the growth medium (Table 2). Compared to the initial value, $\delta^{15}$N values of all *Daphnia* were depleted in $^{15}$N at the end of the experiment in all treatments. The highest $^{15}$N depletion was found at the lowest DIN concentration (0 mg L$^{-1}$ = 3 ‰) and the lowest $^{15}$N depletion was found at the highest DIN concentration (28 mg L$^{-1}$ = 0.2 ‰). With increasing DIN concentrations, $\delta^{15}$N equilibrium values of *Daphnia* increased incrementally from 3.7 ‰ (0 mg L$^{-1}$ DIN) to 8.4 ‰ (28 mg L$^{-1}$ DIN). Starving

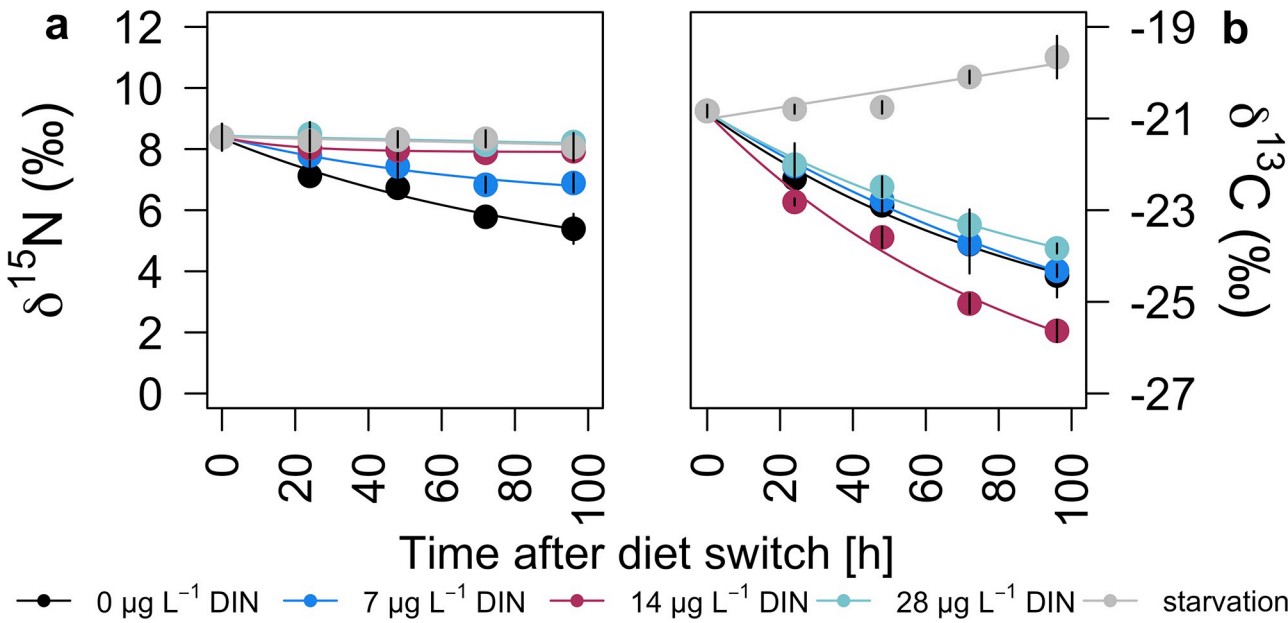

**Fig 4.** Nitrogen (**a**) and carbon (**b**) isotope change over time (96 hour) in *Daphnia* after the diet switch to cyanobacteria and during starvation. Colored lines represent exponential model fits to the different DIN treatments (0, 7, 14, and 28 mg L$^{-1}$ DIN). Linear models were applied to the starvation treatment in **a** and **b** and to the 28 mg L$^{-1}$ DIN treatment in **a**.

**Table 2. Parameter estimates and standard errors of the exponential decay function fitted to the changing $\delta^{15}$N values of *Daphnia* while feeding on the differently grown cyanobacteria for 96 h (n = 3).** $\delta_0$ is the initial isotope ratio, $\delta_{eq}$ the asymptote (plateau), $\lambda$ the incorporation rate of $\delta^{15}$N, AIC is the Akaike Information Criterion, and $T_{50}$ is the half-life isotope turnover rate. For the highest DIN concentration, $\lambda$ and $T_{50}$ were not calculated because of the lack of differences between $\delta_0$ and $\delta_{eq}$.

| DIN (mg L$^{-1}$) | $\delta_0$ (SE) | $\delta_{eq}$ (SE) | log $\lambda$ (SE) | $\lambda$ | AIC | $T_{50}$ [h] |
|---|---|---|---|---|---|---|
| 0 | 8.4 (0.18) | 3.7 (1.90) | -4.55 (0.65) | 0.011 | 17.72 | 65.4 |
| 7 | 8.4 (0.18) | 6.5 (0.65) | -3.99 (0.71) | 0.019 | 16.67 | 37.34 |
| 14 | 8.4 (0.16) | 7.9 (0.15) | -2.91 (1.33) | 0.054 | 12.80 | 12.77 |
| 28 | 8.5 (5e+04) | 8.4 (0.13) | - | - | - | - |

**Table 3. Parameter estimates and standard errors of exponential decay function fitted to the changing $\delta^{13}C$ values of *Daphnia* while feeding on the differently grown cyanobacteria for 96 h (n = 3).** $\delta_0$ is the initial isotope ratio, $\delta_{eq}$ the asymptote (plateau), $\lambda$ the incorporation rate of $\delta^{13}C$, AIC is the Akaike Information Criterion, and $T_{50}$ is the half-life isotope turnover rate.

| DIN (mg L$^{-1}$) | $\delta_0$ (SE) | $\delta_{eq}$ (SE) | log $\lambda$ (SE) | $\lambda$ | AIC | $T_{50}$ [h] |
|---|---|---|---|---|---|---|
| 0 | -20.9 (0.13) | -26.2 (1.28) | -4.51 (0.38) | 0.011 | 8.15 | 62.78 |
| 7 | -20.8 (0.18) | -26.8 (2.39) | -4.67 (0.61) | 0.009 | 16.51 | 75.21 |
| 14 | -20.9 (0.13) | -27.8 (0.98) | -4.40 (0.24) | 0.012 | 6.61 | 56.23 |
| 28 | -20.9 (0.10) | -25.8 (1.30) | -4.66 (0.40) | 0.009 | 1.31 | 73.21 |

animals showed similar $\delta^{15}N$ equilibrium values (8.4 ‰) to animals fed *T. variabilis* cultured at the highest DIN concentrations (Table 2).

$\delta^{13}C$ values of *Daphnia* continuously decreased over time after the diet switch to cyanobacteria in all DIN treatments; most pronounced at 14 mg L$^{-1}$ DIN, resulting in the fastest carbon isotope turn-over rate at this DIN concentration (Fig 4b; Table 3). In animals starving after the diet switch at day 5 of the experiment, $\delta^{13}C$ values increased over time while $\delta^{15}N$ values remained constant (Fig 4a and 4b).

## Elemental composition of *Daphnia*

After the diet switch at day 5, the relative N content (%) of the animals decreased over time until the end of the experiment (t$_0$-t$_4$), significantly in the starvation treatment (3.6%; Tukey's HSD, $p = 0.004$) and slightly in the various DIN treatments (Fig 5a). The relative N content of starving animals differed significantly from the relative N content of animals reared on the different cyanobacterial DIN treatments (ANOVA, $F_{5,15} = 17.41$, $p < 0.001$. The relative C content (%) of the animals also decreased over time in most treatments, significantly in the starvation treatment (14.1%; ANOVA, $F_{5,15} = 13.17$, $p < 0.001$) but less pronounced and non-uniformly in the various DIN treatments (Fig 5b). The molar C:N ratio of starving animals and that of animals of the lowest cyanobacterial DIN treatment (0 mg L$^{-1}$) increased over time, whereas the molar C:N ratio of animals of all other DIN treatments did not change significantly (Fig 5c; ANOVA, $F_{5,15} = 5.07$, $p = 0.006$).

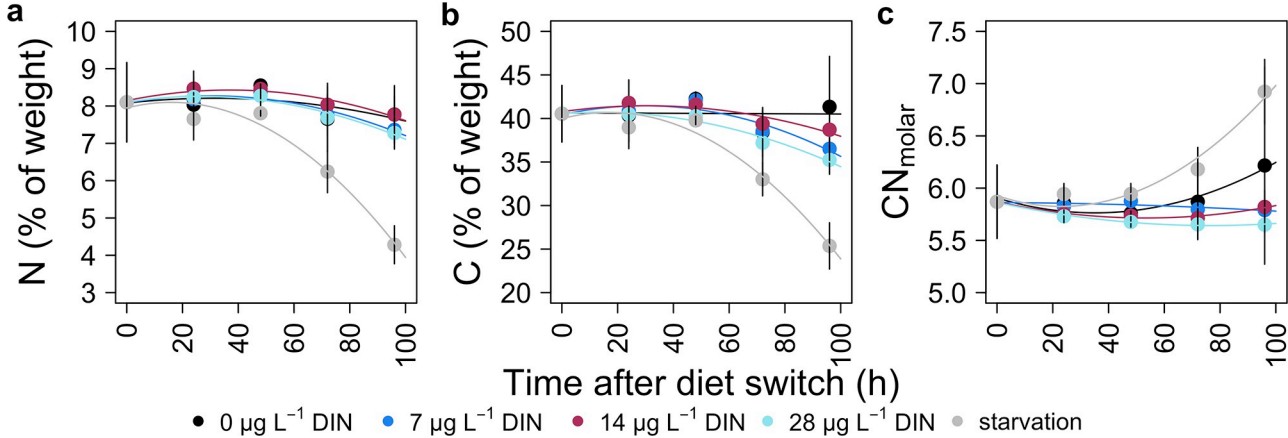

**Fig 5.** Change in relative nitrogen (**a**) and carbon (**b**) content (% of dry weight) as well as molar C:N ratios of Daphnia after the diet switch to cyanobacteria at day 5 of the experiment and during starvation. Colored lines represent exponential model fits to the different DIN treatments (0, 7, 14, and 28 mg L$^{-1}$ DIN).

The $\delta^{15}$N values of *Daphnia* reared on the different DIN treatments did not correlate with body N and C content (%) or molar C:N ratio (Pearson: $p > 0.05$). In contrast, $\delta^{15}$N values of starving animals were positively correlated with N (Pearson: $r = 0.63$, $p = 0.012$) and C (Pearson: $r = 0.61$, $p = 0.017$) and negatively with the molar C:N ratio (Pearson: $r = -0.67$, $p = 0.007$). The $\delta^{13}$C values of *Daphnia* of the highest DIN treatment (28 mg L$^{-1}$) were positively correlated with relative body C content (Pearson: $r = 0.57$, $p = 0.02$) and molar C:N ratio (Pearson: $r = 0.51$, $p = 0.044$). In all other DIN treatments, the $\delta^{13}$C values of *Daphnia* did not correlate significantly with either body N or C content or C:N ratio. In contrast, $\delta^{13}$C of starving animals were negatively correlated with body N (Pearson: $r = -0.90$, $p < 0.001$) and C content (Pearson: $r = -0.92$, $p < 0.001$) and positively with C:N ratios (Pearson: $r = 0.83$, $p < 0.001$).

### Trophic fractionation

In general, $\delta^{15}$N of daphnids of the different DIN treatments were enriched by 5.1 ‰ ($\pm$ 0.6) compared to their diet (Fig 6a). The $\Delta^{15}$N$_{\text{cons-diet}}$ values were non-linearly correlated with the DIN concentration in the cyanobacterial culture medium (Pearson: $r = 0.91$, $p = 0.03$); the values first decreased with increasing DIN concentration until 14 mg L$^{-1}$ and then increased again at 28 mg L$^{-1}$ (Fig 6a). The $^{15}$N trophic enrichment was thus highest (5.8 ‰) in the absence of DIN in the cyanobacterial culture medium. The $\Delta^{15}$N$_{\text{cons-diet}}$ values of *Daphnia* showed an inverse relationship with the nitrogen content of their diet, but this was not significant (Pearson: $r = -0.55$, $p = 0.053$; Fig 6c). Additionally, there was no relationship between the $\Delta^{15}$N$_{\text{cons-diet}}$ values of *Daphnia* and the molar C:N ratios of their diet (Pearson: $r = 0.11$, $p > 0.05$; Fig 6c). However, when considering the trophic N fractionation ($\Delta^{15}$N$_{\text{cons-diet}}$) in comparison to the elemental composition of the animals, there was a significant negative correlation with the molar C:N body ratio of the animals (Pearson: $r = -0.60$, $p = 0.037$; Fig 6i). We did not find any relationship between trophic fractionation of N and body N content (%) (Fig 6g).

The $\Delta^{13}$C$_{\text{cons-diet}}$ values increased with increasing DIN availability in the cyanobacterial culture medium (Pearson: $r = 0.8$, $p < 0.001$; Fig 6b); the trophic enrichment thus increased from 3.3 ‰ at 0 mg L$^{-1}$ DIN to 4.2 ‰ at 28 mg L$^{-1}$ DIN. The $\Delta^{13}$C$_{\text{cons-diet}}$ values of *Daphnia* showed a positive trend with the carbon content of their diet, which was not significant (Pearson: $r = -0.55$, $p = 0.052$; Fig 6d). The molar C:N ratios of the diet did not correlate with the $\Delta^{13}$C$_{\text{cons-diet}}$ values of *Daphnia* (Pearson: $r = 0.15$, $p > 0.05$; Fig 6f). However, when considering the trophic C fractionation ($\Delta^{13}$C$_{\text{cons-diet}}$) of *Daphnia* in comparison to the elemental composition of the *Daphnia*, there was no significant relationship to the C content (Pearson: $r = -0.23$, $p > 0.05$; Fig 6d) and the molar C:N ratio (Pearson: $r = -0.07$, $p > 0.05$; Fig 6f).

### Body dry weight of *D. magna*

The dry weight of the animals did not change significantly during the experimental exposure to the cyanobacteria (from t$_0$ to t$_4$) in none of the DIN treatments and the observed dry weight changes did not differ among treatments (ANOVA, $F_{4,11} = 2.028$, $p > 0.05$). The starving animals lost weight from t$_0$ to t$_4$ (on average 49.9% of the initial body dry weight at t$_0$).

### Discussion

In our experiment, the $\delta^{15}$N values of the diazotrophic cyanobacterium *T. variabilis* were positively correlated with the availability of DIN in the culture medium, suggesting a successive change in N acquisition from N$_2$ fixation at low DIN concentrations to the uptake of DIN at high DIN concentrations. In other words, the increasing depletion of $\delta^{15}$N from 3.1 to -0.7 ‰ with decreasing DIN concentrations likely reflects increasing N$_2$ fixation to compensate for

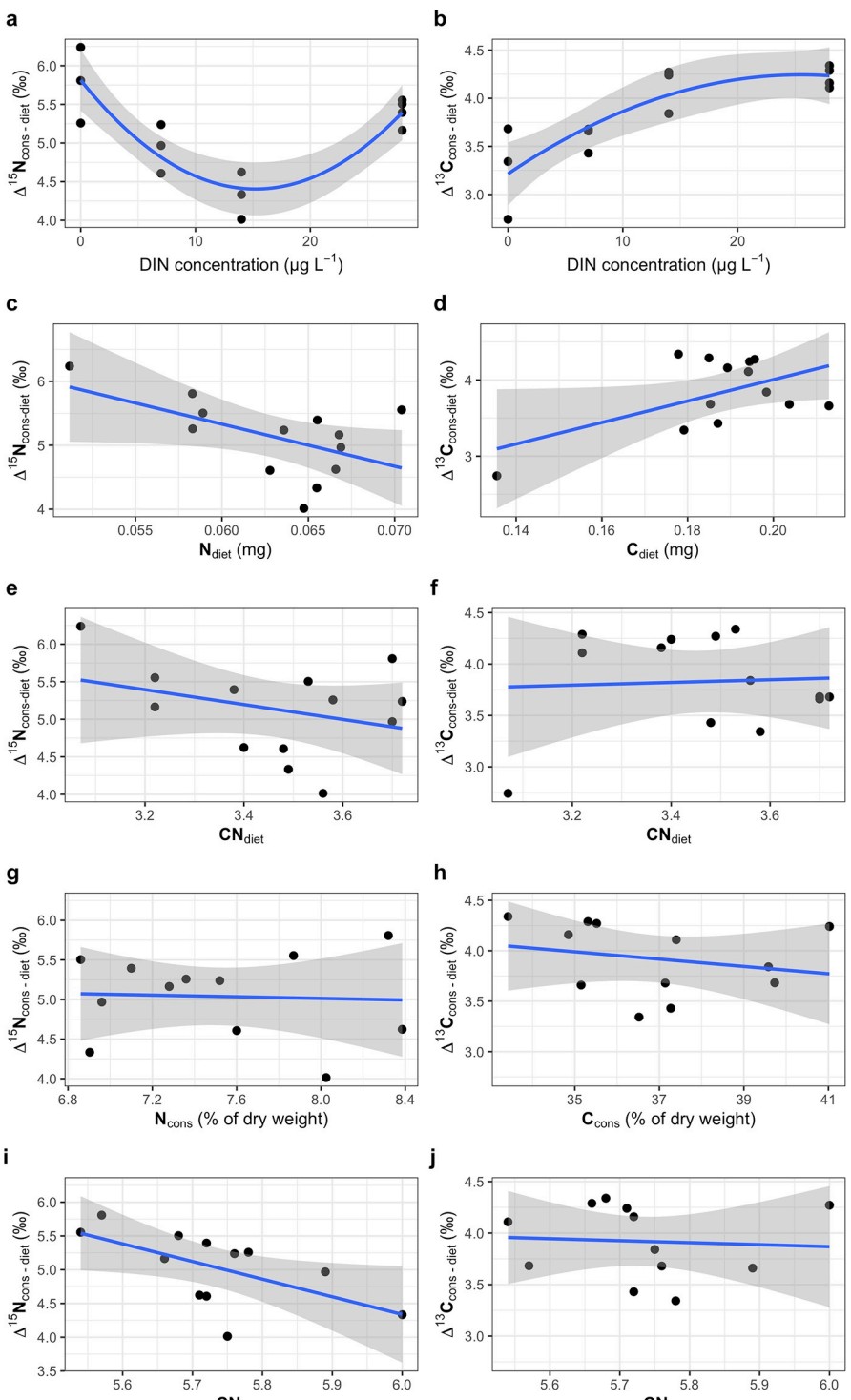

**Fig 6.** Trophic fractionation of $^{15}N$ and $^{13}C$. $\Delta^{15}N_{cons\text{-}diet}$ and $\Delta^{13}C_{cons\text{-}diet}$ values versus DIN concentration in the cyanobacterial culture medium (**a** and **b**), N and C content of the diet (**c** and **d**), molar C:N ratio of the diet (**e** and **f**), body N and C content of *Daphnia* (% of dry weight; **g** and **h**), and molar C:N ratios of *Daphnia* (**i** and **j**).

decreasing DIN availability. The strong depletion in $\delta^{15}N$ values in the absence of DIN corroborates previous findings. Bauersachs et al. [17] reported even lower values of $\delta^{15}N$ (-1.85 ‰) for the same strain of *T. variabilis* cultivated with $N_2$ gas. A depletion in $\delta^{15}N$ values (up to -0.5 ‰) at low DIN concentrations was also observed in field studies conducted in both eutrophic and oligotrophic lakes [44–46]. *Anabaena variabilis* has been reported to discriminate against $^{15}N$ during N fixation, resulting in $\delta^{15}N$ values of approximately -1 ‰ [18, 19]. The non-linear relationship we observed between $\delta^{15}N$ and DIN availability suggests a switch to $N_2$ fixation at a DIN concentration of $<14$ mg $L^{-1}$ ($NaNO_3 <85$ mg $L^{-1}$). Based on bottle incubation experiments conducted in P-rich reservoirs, Bradburn et al. [47] concluded that the efficiency of $N_2$ fixation increases with decreasing DIN availability and is highest at DIN concentrations of $<50$ μg $L^{-1}$ and low irradiance (due to photoinhibition). In a mesocosm study, $N_2$ fixation has been found to increase with decreasing N:P [12]. Whether this translates to lower thresholds for $N_2$ fixation in oligotrophic lakes remains unclear. The relationship between nutrient loading, productivity and biological N transformations ($N_2$ fixation versus denitrification) is complex [48] and requires further investigation, also in regard to environmental thresholds [49] potentially allowing to assess the onset of $N_2$ fixation and to explore seasonal changes in the balance between biological N transformations.

The molar C:N ratios of *T. variabilis* cells did not differ significantly among the different DIN treatments, but tended to correlate negatively with increasing DIN availability (not considering the C:N ratio at 0 mg $L^{-1}$ DIN; Fig 1C). The small peak in C:N ratios at 7 mg $L^{-1}$ DIN may reflect a stoichiometric response to the switch to $N_2$ fixation. Overall, however, C:N ratios of *T. variabilis* were consistently low, ranging from 3.3 to 3.6 (mean values), suggesting that *T. variabilis* did not experience N limitation in any of the treatments. Thus, the switch to $N_2$ fixation presumably allowed *T. variabilis* to overcome N limitation at low DIN concentrations. This also indicates that *T. variabilis* switches to energetically more expensive $N_2$ fixation only if DIN becomes limiting [9–11, 50]. Stoichiometric N homeostasis has been reported already from other diazotrophic cyanobacteria. In *Dolichospermum*, stoichiometric N homeostasis has been found to come at the expense of reduced biomass production during $N_2$ fixation [13, 51]. In contrast, *Aphanizomenon flos-aquae* has been found to exhibit flexible C:N stoichiometry and no growth trade-off due to N-fixation, potentially reflecting a high diversity in strategies to cope with N deficiencies among diazotrophs [52]. In our experiment, biomass production (i.e., carbon concentrations at the end of the experiment) did not differ among DIN treatments, suggesting that *T. variabilis* is able to maintain stoichiometric N homeostasis during $N_2$ fixation without compromising growth.

Nutrient limitation, i.e., N limitation [31, 34] and P limitation [32, 33, 35], as well as the relative availability of $CO_2$ and $HCO_3^-$ [34] can also influence the C isotope signature of photosynthetic organisms through changes in the activity of the $CO_2$ concentrating mechanism (CCM). In our experiment, the $\delta^{13}C$ values of *T. variabilis* did not show a clear relationship with DIN availability (Fig 1B). The 2.1 ‰ depletion in $\delta^{13}C$ values at a DIN concentration of 14 mg $L^{-1}$ ($\delta^{13}C$ = -29.7 ‰ compared to an average of -27.6 ‰ in the other treatments) may be related to changes in C acquisition. However, our cyanobacteria cultures were consistently aerated ($CO_2$ supply) and additionally supplied with $HCO_3^-$ via the growth medium, making C limitation rather unlikely. Thus, the switch to $N_2$ fixation probably had no effect on the uptake of carbon in our experiment.

Field studies suggest that the stable isotope signature of consumers can reflect the assimilation of organic matter produced by N-limited, diazotrophic cyanobacteria [24, 26–29]. However, this has rarely been explored under controlled laboratory conditions [26]. We show here experimentally that the mode of N acquisition at the primary producer level (i.e., $N_2$ fixation versus DIN uptake) not only affects the stable isotope values of the primary producers but also

those of their consumers (i.e., *Daphnia*). $\delta^{15}$N values of *Daphnia* decreased over time while feeding on the differently grown cyanobacteria, most evident at DIN concentrations $\leq$7 mg $L^{-1}$ in the growth medium (Fig 2A). After 96 h of feeding, the highest $^{15}$N depletion was found in *Daphnia* in the absence of DIN (0 mg $L^{-1}$ = 3 ‰) and the lowest $^{15}$N depletion was found in *Daphnia* at the highest DIN concentration (28 mg $L^{-1}$ = 0.2 ‰) in the cyanobacterial growth medium, reflecting the switch from $N_2$ fixation to DIN uptake at the primary producer level. The $\delta^{15}$N values of *Daphnia* were positively correlated with the $\delta^{15}$N values of *T. variabilis*, i.e., $N_2$ fixation was clearly detectable also at the consumer level. The N isotope turnover rates, expressed as the time it takes for 50% of the stable isotopes in a consumer's tissue to be replaced by the stable isotopes in the diet ($T_{50}$ values), decreased with increasing DIN concentration in the growth medium. Thus, $T_{50}$ values of *Daphnia* were highest in the absence of DIN as a consequence of $N_2$ fixation and resulting depletion in $^{15}$N at the primary producer level. The larger the isotope differences between consumer and diet, the longer it takes until the isotope signature of the consumer reflects that of its diet. In our setting, it took 65.4 h until 50% of the stable isotopes in *Daphnia* tissue were replaced by stable isotopes derived from *T. variabilis* acquiring its N via $N_2$ fixation. The $^{15}$N depletion caused by $N_2$ fixation at the primary producer level should thus be detectable also at the consumer level (zooplankton) within a few days (2–3) after the switch to $N_2$ fixation. The $\delta^{15}$N values of *Daphnia* (96 h after diet switch) increased asymptotically with DIN concentration in the cyanobacterial growth medium, also reflecting the switch from $N_2$ fixation to DIN uptake at the primary producer level and the establishment of consumer-food isotope equilibria.

The $\Delta^{15}$N$_{cons-diet}$ values of *Daphnia* showed a U-shaped relationship with DIN concentration in the cyanobacterial growth medium, suggesting that $^{15}$N trophic fractionation changed non-linearly with DIN availability–a finding that requires further investigations. On average (i.e., across DIN treatments), *Daphnia* were enriched in $^{15}$N by 5.1 ‰ (±0.6 ‰, $\Delta^{15}$N$_{cons-diet}$) compared to its diet. $\Delta^{15}$N values are known to vary considerably across trophic levels, because most consumers feed on a variety of different food sources and often in varying proportions [53, 54]. Adams and Sterner [55] showed already that even within a consumer-resource pair (*Daphnia*-green algae), strong differences in $\Delta^{15}$N values from 1 to 6 ‰ can occur due to differences in N content of the diet. The strong positive relationship between $^{15}$N enrichment ($\Delta^{15}$N$_{cons-diet}$) and C:N ratio of the algal diet reported by Adams and Sterner [55] was not expected in our study because we used a diazotrophic cyanobacterium as food, capable of switching to $N_2$ fixation at low DIN concentrations, thus avoiding N limitation. The $\delta^{15}$N values of the cyanobacterium increased significantly with the switch from $N_2$ fixation to DIN uptake, potentially also affecting the trophic enrichment of $^{15}$N. Adams and Sterner [55] reported that the $\delta^{15}$N values of *Daphnia* and the consumer-diet isotope fractionation factor ($\Delta^{15}$N$_{cons-diet}$) were inversely related to the N content of the algae. We found a significant relationship between ($\Delta^{15}$N$_{cons-diet}$) and C:N body ratio of the animals, but no relationship with the C:N ratio or N content of their cyanobacterial diet.

The carbon and nitrogen content of *Daphnia* exposed to the cyanobacterium *T. variabilis* for four days decreased slightly, but much less than in the starvation treatment, indicating that the animals were able to ingest the *T. variabilis* filaments and maintain their carbon and nitrogen content by consuming *T. variabilis*. Cyanobacteria are unsuitable as food for *Daphnia* and other invertebrate consumers for a number of reasons, including their potential toxicity and the lack of essential lipids [56–58]. *Daphnia* feeding on *T. variabilis* (ATCC 29413, formerly *Anabaena variabilis*) have been shown to struggle primarily with sterol limitation, resulting in very low somatic growth rates [57, 59]. Also in our experiment, the dry mass of *Daphnia* feeding on *T. variabilis* increased only slightly. However, the stable isotope data showed that the animals incorporated and processed the carbon and nitrogen provided by the cyanobacterium,

suggesting that the animals were able to at least meet their basic energy requirements by consuming the cyanobacterium. $\delta^{15}$N and $\delta^{13}$C values of consumers can change due to food quantity- and quality-related isotope fractionation [60]. During periods of nutritional stress, enrichment in $^{15}$N may result from the segregation of $^{14}$N, i.e., the excretion of isotopically light urea [61, 62]. In the present study, $\delta^{15}$N values of starving *Daphnia* decreased only slightly (by 0.5 ‰), whereas the N weight fraction (% of body weight) strongly decreased (by 43.3%), suggesting similar excretion of $\delta^{15}$N and $\delta^{14}$N during starvation. Doi et al. [63] conducted a meta-analysis to evaluate the effects of starvation on $\delta^{15}$N of consumers and found that the $\delta^{15}$N values of most consumers increased with the length of starvation. The four-day starvation period applied in our study might have been too short to affect the $\delta^{15}$N values of *Daphnia*.

In contrast to $\delta^{15}$N values, the $\delta^{13}$C values of starving *Daphnia* increased over time, which has been reported previously [60, 64]. Enrichment in $^{13}$C during starvation may result from increased metabolization of $^{13}$C-depleted lipids in order to meet energetic demands [63]. The samples obtained in our study were not subjected to lipid extraction prior to stable isotope analysis. After the diet switch to cyanobacteria, $\delta^{13}$C values of *Daphnia* continuously decreased in all DIN treatments, with no clear relationship with DIN concentration. Decreasing $\delta^{13}$C values after a diet switch to cyanobacteria have been reported previously [60]. One possible explanation could be that the animals incorporated $^{13}$C-depleted lipids from the cyanobacterial food but were unable to grow and reproduce due to sterol limitation and thus did not further metabolize the accumulating lipids and also could not allocate them to reproduction. At high food quantity and low dietary sterol supply (as in the present study), *Daphnia* have been reported to increase excretion and respiration of excess bulk C [65], which should lead to an enrichment in bulk $^{13}$C, i.e., increasing $\delta^{13}$C values. Decreasing $\delta^{13}$C values in consumers in response to declining food quality should therefore be further investigated to elucidate the underlying mechanisms. The $^{13}$C enrichment ($\Delta^{13}C_{cons-diet}$) in *Daphnia* was in the range of 3.1 to 4.1 ‰ compared to its cyanobacterial diet, and thus much larger than the commonly assumed trophic enrichment of $\leq 1$ ‰ for C [54, 66, 67], potentially also reflecting nutritional constraints. The C stable isotope data at the consumer level generally did not show a clear relationship with the DIN concentration provided at the primary producer level.

## Conclusion

We show here experimentally that the $\delta^{15}$N values of the diazotrophic cyanobacterium *T. variabilis* are positively related to the concentration of DIN in the growth medium, indicating a switch from $N_2$ fixation to DIN uptake with increasing DIN availability. Thus, $N_2$ fixation occurs only at low and potentially limiting DIN concentrations ($<14$ mg $L^{-1}$), most likely because $N_2$ fixation is energetically more costly than DIN uptake. C:N ratios of *T. variabilis* were consistently low, suggesting that $N_2$ fixation allowed *T. variabilis* to overcome N limitation at low DIN concentrations. The $\delta^{13}$C values of *T. variabilis* did not show a clear relationship with DIN availability. The mode of N acquisition at the primary producer level also influenced the stable isotope values of the consumer *Daphnia*. The $\delta^{15}$N values of *T. variabilis* were positively correlated with the $\delta^{15}$N values of *Daphnia* and the estimated N isotope turnover rates suggest that a switch to $N_2$ fixation at the primary producer level should be detectable at the consumer level within a few days (2–3). The diet switch to cyanobacteria strongly influenced the $\delta^{15}$N and $\delta^{13}$C values of *Daphnia* due to food quality-related isotope fractionation, but this did not affect the detectability of diazotrophic N in the consumer. We conclude that stable isotope analysis of bulk N at the phytoplankton-zooplankton interface can provide valuable information about N limitation and the fate of diazotrophic N in lake food webs.

## Author Contributions

**Conceptualization:** Michelle Helmer, Desiree Helmer, Elizabeth Yohannes, Dominik Martin-Creuzburg.

**Data curation:** Michelle Helmer.

**Formal analysis:** Michelle Helmer.

**Funding acquisition:** Elizabeth Yohannes, Daniel R. Dietrich, Dominik Martin-Creuzburg.

**Investigation:** Michelle Helmer, Desiree Helmer, Jason Newton.

**Methodology:** Michelle Helmer, Desiree Helmer, Elizabeth Yohannes, Jason Newton.

**Project administration:** Michelle Helmer, Desiree Helmer.

**Resources:** Jason Newton.

**Software:** Michelle Helmer.

**Supervision:** Elizabeth Yohannes, Daniel R. Dietrich, Dominik Martin-Creuzburg.

**Visualization:** Michelle Helmer.

**Writing – original draft:** Michelle Helmer, Dominik Martin-Creuzburg.

**Writing – review & editing:** Michelle Helmer, Jason Newton, Daniel R. Dietrich, Dominik Martin-Creuzburg.

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
