## [Decision Letter · Decision Letter 0]

27 May 2024

PONE-D-24-09860Dissolved nitrogen uptake versus nitrogen fixation: Mode of nitrogen acquisition affects stable isotope signatures of a diazotrophic cyanobacterium and its grazerPLOS ONE

Dear Dr. Martin-Creuzburg,

Thank you for submitting your manuscript to PLOS ONE. I want to start by apologizing for the length of time it took to have your manuscript reviewed. I contacted a number of potential reviewers, but it took a significant amount of time to identify two reviewers who were willing and able to review the manuscript. Having said that, these two reviewers judged that your manuscript will be suitable for publication after some minor changes are made. Both reviewers mentioned the number of significant units you are reporting stable isotope data for, meaning that the resolution reported is better than the analytical precision. Another joint comment was about the nitrogen isotope composition of nitrate often being below zero.  Reviewer #2 also had a specific concern regarding the turnover time it takes to replace half of Daphnia’s tissue N and that it should not depend on the magnitude of difference in stable isotope signatures between consumer and food source. (Point #1). Both reviewers also made critical comments regarding Figure 3. I want to take this opportunity, therefore to invite you to submit a revised version of the manuscript that addresses the points raised during the review process.

We look forward to receiving your revised manuscript.

Kind regards,

Lee W Cooper, Ph.D.

Section Editor

PLOS ONE

Journal Requirements:

"The work was financially supported by the Ministry of Science, Research, and the Arts of the  Federal State Baden-Württemberg, Germany (Water Research Network Project: Challenges of  Reservoir Management—Meeting Environmental and Social Requirements) and by the Deutsche Forschungsgemeinschaft (DFG, German Research Foundation, 298726046/GRK2272). The authors declare no conflict of interest. The funders had no role in the design of the study; in  the collection, analyses, or interpretation of data; in the writing of the manuscript; or in the decision to publish the results."

"The work was financially supported by the Ministry of Science, Research, and the Arts of the Federal State Baden-Württemberg, Germany (Water Research Network Project: Challenges of Reservoir Management—Meeting Environmental and Social Requirements, to D.M.-C. and D.D.) and by the Deutsche Forschungsgemeinschaft (DFG, German Research Foundation, 298726046/GRK2272, to D.M.-C. and E.Y.). The authors declare no conflict of interest. The funders had no role in the design of the study; in the collection, analyses, or interpretation of data; in the writing of the manuscript; or in the decision to publish the results."

4. We notice that your supplementary tables are included in the manuscript file. Please remove them and upload them with the file type 'Supporting Information'. Please ensure that each Supporting Information file has a legend listed in the manuscript after the references list.

Reviewers' comments:

Reviewer's Responses to Questions

**Comments to the Author**

1. Is the manuscript technically sound, and do the data support the conclusions?

Reviewer #1: Yes

Reviewer #2: Partly

2. Has the statistical analysis been performed appropriately and rigorously? 

Reviewer #1: Yes

Reviewer #2: Yes

3. Have the authors made all data underlying the findings in their manuscript fully available?

Reviewer #1: Yes

Reviewer #2: No

4. Is the manuscript presented in an intelligible fashion and written in standard English?

Reviewer #1: Yes

Reviewer #2: Yes

5. Review Comments to the Author

Reviewer #1: This is an interesting paper that details variations in the nitrogen isotope composition of cyanobacteria and Daphnia in controlled experiments that varied concentrations of nitrate, which provides insights on nitrate utilization relative to organic nitrogen generated through nitrogen fixation. I don’t think the paper explicitly proves that the nitrate starved experiments resulted in increases in nitrogen fixation, but these seem like reasonable assumptions, and perhaps a cleverly designed enriched isotope experiment would be of value as a follow-up. The changes observed in the feeding experiments are relatively small, a few per mil, but have consequences for understanding variation in nitrogen isotope ratios within foodwebs. The non-linear change observed in nitrogen isotope fractionation relative to nitrate concentrations is a particularly interesting finding. There is also an interesting investigation of organic material turnover time in Daphnia that could be emphasized more in the abstract. The discussion is particularly well-written and provides good context for the data presented by citing a number of prior studies.

Line 61. I am not sure I would say that 15N is an “excellent” tracer---nitrogen cycling is complex from the standpoint of isotopic fractionation. On the next line it is stated that diatomic nitrogen gas is isotopically depleted in 15N relative to nitrate, but this isn’t universally true. Atmospheric nitrate is commonly depleted in 15N relative to N2, so if nitrate in aqueous sources has been in part derived from atmospheric deposition or as a result of nitrification in soils, it is possible that the nitrate will have negative delta values relative to N2 gas. One recent paper that shows this is Matiatos, I., Wassenaar, L.I., Monteiro, L.R. et al. Global patterns of nitrate isotope composition in rivers and adjacent aquifers reveal reactive nitrogen cascading. Commun Earth Environ 2, 52 (2021). https://doi.org/10.1038/s43247-021-00121-x

Line 67-68. Again, the full range of nitrogen isotope composition of nitrate is not acknowledged here. The paper cited, Soto et al. 2019, is a study of a single lake basin in Canada, and doesn’t provide a global overview of the full range of nitrate isotopic compositions possible. I don’t think it makes the follow-on statements untrue about the potential use of 15N as a tracer, but the introduction doesn’t convey the full complexity of isotopic transformations that are possible in the nitrogen cycle.

Line 149. Isotope masses on the delta values need to be superscripted. Was the isotopic composition of the sodium nitrate used in the study measured, and was it homogeneous?

Line 209. Since the relationship between the nitrate concentration and the nitrogen isotope composition is not linear (Figure 1a), is it appropriate to report a Pearson correlation coefficient? Doesn’t that reflect a linear correlation that clearly isn’t present here? Line 211. Since the reported precision for the nitrogen isotope measurements is 0.15 per mil, reporting data to two significant digits, e.g. -0.72 is not consistent with that reported precision. The data reported throughout the manuscript are reported to two decimal places.

Figure 3b. Since there isn’t an obvious correlation between DIN and the carbon isotope composition, the sinusoidal function that is plotted on the figure doesn’t really have any meaning.

Line 258 Treatment should be treatments

Table 3. Again, as with the nitrogen isotope data, since the precision is 0.09 per mil, (~0.1 per mil), it is dubious to use two decimal places to report the data.

Some editing of the bibliography would be of benefit. For example, superscripts and subscripts where needed, e.g. Baker et al. 2018, Fry and Sherr, 1989, Loick-Wilde et al. 2012, McClelland et al. 2003, Oelbermann and Scheu, 2002; italicization of genus and species names, e.g. Bradburn et al. 2012; capitalization of proper nouns, e.g. Ambrose and DeNiro reference; inconsistent capitalization of all words in an article title, e.g. Karlson et al. 2014, Meeks et al. 1983; more bibliographic information needed for an edited book, e.g. Montoya, 2008.

Reviewer #2: This manuscript presents experimental results showing how the freshwater diazotroph Trichormus variabilis switches from N-fixation to use of nitrate when cultured with increasing nitrate concentration. Changes in the stable N isotope ratios of Trichormus in different nitrate treatments were used to demonstrate the switch. Stable isotope analyses were also used to show how changes in stable N isotope values in Trichormus are passed on to a consumer (Daphnia). Stable carbon isotope ratios were also measured in this study, though the focus is primarily on nitrogen. The experiments present in this paper were well designed and well executed, and the resultant data are instructive. The results largely support expectations based on previous work, but controlled laboratory studies of N switching by diazotrophs and stable isotopic relationships between diazotrophs and their consumers are nonetheless useful as a basis for refining our ability to interpret field observations. Some suggestions to improve the manuscript are provided below.

1) While most of the findings make sense, I’m a bit perplexed by the findings related to stable N isotope equilibration rates between Daphnia and Trichormus in the different treatments. The time it takes to replace half of Daphnia’s tissue N with the new food source signature should not depend on the magnitude of difference in stable isotope signatures between the consumer and its food source. Are you suggesting that somehow N from Trichormus grown with increasing amounts of nitrate becomes more easily assimilated by Daphnia? Or is this result related to the precision of the isotope measurements? The smaller the difference in isotopic value between food source and consumer, the more uncertain the estimate of tissue turnover time becomes (the error surrounding the measurements becomes an increasing proportion of the difference between them). In the case of Trichormus grown with 28 mg/L DIN (Table 2, bottom row), the initial and equilibrated isotope values are essentially the same. How can an equilibration time even be calculated when this is the case? This part of the manuscript needs some clarification.

2) Throughout paper – Reporting results to two places after the decimal point indicates a level of precision that is not supported. Rounding to one place after the decimal would be more appropriate.

3) Lines 61-62. The blanket statement that N2 is depleted in 15N compared to nitrate is not correct. Stable N isotope values in rainwater nitrate are often below zero.

4) I suggest deleting Figure 3. Since any potential linkage between Daphnia and DIN concentration is via their consumption of Trichormus, Figure 2 should is sufficient. Plotting Daphnia values versus DIN concentrations suggests a direct linkage that does not exist.

6. PLOS authors have the option to publish the peer review history of their article (what does this mean?). If published, this will include your full peer review and any attached files.

Reviewer #1: No

Reviewer #2: No

---

## [Author Response · Author response to Decision Letter 0]

7 Jun 2024

Response to Reviewers – Helmer et al.

Reviewer #1: This is an interesting paper that details variations in the nitrogen isotope composition of cyanobacteria and Daphnia in controlled experiments that varied concentrations of nitrate, which provides insights on nitrate utilization relative to organic nitrogen generated through nitrogen fixation. I don’t think the paper explicitly proves that the nitrate starved experiments resulted in increases in nitrogen fixation, but these seem like reasonable assumptions, and perhaps a cleverly designed enriched isotope experiment would be of value as a follow-up. The changes observed in the feeding experiments are relatively small, a few per mil, but have consequences for understanding variation in nitrogen isotope ratios within food webs. The non-linear change observed in nitrogen isotope fractionation relative to nitrate concentrations is a particularly interesting finding. There is also an interesting investigation of organic material turnover time in Daphnia that could be emphasized more in the abstract. The discussion is particularly well-written and provides good context for the data presented by citing a number of prior studies.

Response: Thank you for this very positive feedback. As suggested, we now emphasize our results on nitrogen turnover rates already in the abstract by adding: ‘Nitrogen isotope turnover rates of Daphnia were highest in the absence of DIN as a consequence of N2 fixation and resulting depletion in 15N at the primary producer level.’

Line 61. I am not sure I would say that 15N is an “excellent” tracer---nitrogen cycling is complex from the standpoint of isotopic fractionation. On the next line it is stated that diatomic nitrogen gas is isotopically depleted in 15N relative to nitrate, but this isn’t universally true. Atmospheric nitrate is commonly depleted in 15N relative to N2, so if nitrate in aqueous sources has been in part derived from atmospheric deposition or as a result of nitrification in soils, it is possible that the nitrate will have negative delta values relative to N2 gas. One recent paper that shows this is Matiatos, I., Wassenaar, L.I., Monteiro, L.R. et al. Global patterns of nitrate isotope composition in rivers and adjacent aquifers reveal reactive nitrogen cascading. Commun Earth Environ 2, 52 (2021). https://doi.org/10.1038/s43247-021-00121-x

Response: Thank you for pointing out the interesting paper, which we are now also citing and thus highlighting the complexity of possible isotopic fractionations that are happening in the nitrogen cycle.

Line 67-68. Again, the full range of nitrogen isotope composition of nitrate is not acknowledged here. The paper cited, Soto et al. 2019, is a study of a single lake basin in Canada, and doesn’t provide a global overview of the full range of nitrate isotopic compositions possible. I don’t think it makes the follow-on statements untrue about the potential use of 15N as a tracer, but the introduction doesn’t convey the full complexity of isotopic transformations that are possible in the nitrogen cycle.

Response: Please, see above

Line 149. Isotope masses on the delta values need to be superscripted. Was the isotopic composition of the sodium nitrate used in the study measured, and was it homogeneous?

Response: Has been changed. Yes, the isotopic composition of sodium nitrate was homogeneous.

Line 209. Since the relationship between the nitrate concentration and the nitrogen isotope composition is not linear (Figure 1a), is it appropriate to report a Pearson correlation coefficient? Doesn’t that reflect a linear correlation that clearly isn’t present here? Line 211. Since the reported precision for the nitrogen isotope measurements is 0.15 per mil, reporting data to two significant digits, e.g. -0.72 is not consistent with that reported precision. The data reported throughout the manuscript are reported to two decimal places.

Response: That’s right. We used the asymptotic regression model to describe the non-linear relationship between DIN and delta15N, as described in the method section. The Pearson statistics was erroneously reported here and has now been removed. Thanks for pointing this out.

We agree with the comment on the reported precision and removed the second decimal place of the delta values throughout the manuscript.

Figure 3b. Since there isn’t an obvious correlation between DIN and the carbon isotope composition, the sinusoidal function that is plotted on the figure doesn’t really have any meaning.

Response: Yes, probably not. We removed the function from the Figure 3b.

Line 258 Treatment should be treatments

Response: Has been changed

Table 3. Again, as with the nitrogen isotope data, since the precision is 0.09 per mil, (~0.1 per mil), it is dubious to use two decimal places to report the data.

Response: Has been changed

Some editing of the bibliography would be of benefit. For example, superscripts and subscripts where needed, e.g. Baker et al. 2018, Fry and Sherr, 1989, Loick-Wilde et al. 2012, McClelland et al. 2003, Oelbermann and Scheu, 2002; italicization of genus and species names, e.g. Bradburn et al. 2012; capitalization of proper nouns, e.g. Ambrose and DeNiro reference; inconsistent capitalization of all words in an article title, e.g. Karlson et al. 2014, Meeks et al. 1983; more bibliographic information needed for an edited book, e.g. Montoya, 2008.

Response: The reference list has been revised

Reviewer #2: This manuscript presents experimental results showing how the freshwater diazotroph Trichormus variabilis switches from N-fixation to use of nitrate when cultured with increasing nitrate concentration. Changes in the stable N isotope ratios of Trichormus in different nitrate treatments were used to demonstrate the switch. Stable isotope analyses were also used to show how changes in stable N isotope values in Trichormus are passed on to a consumer (Daphnia). Stable carbon isotope ratios were also measured in this study, though the focus is primarily on nitrogen. The experiments present in this paper were well designed and well executed, and the resultant data are instructive. The results largely support expectations based on previous work, but controlled laboratory studies of N switching by diazotrophs and stable isotopic relationships between diazotrophs and their consumers are nonetheless useful as a basis for refining our ability to interpret field observations. Some suggestions to improve the manuscript are provided below.

Response: Thank you for this positive feedback.

1) While most of the findings make sense, I’m a bit perplexed by the findings related to stable N isotope equilibration rates between Daphnia and Trichormus in the different treatments. The time it takes to replace half of Daphnia’s tissue N with the new food source signature should not depend on the magnitude of difference in stable isotope signatures between the consumer and its food source. Are you suggesting that somehow N from Trichormus grown with increasing amounts of nitrate becomes more easily assimilated by Daphnia? Or is this result related to the precision of the isotope measurements? The smaller the difference in isotopic value between food source and consumer, the more uncertain the estimate of tissue turnover time becomes (the error surrounding the measurements becomes an increasing proportion of the difference between them). In the case of Trichormus grown with 28 mg/L DIN (Table 2, bottom row), the initial and equilibrated isotope values are essentially the same. How can an equilibration time even be calculated when this is the case? This part of the manuscript needs some clarification.

Response: Well, the switch from N2 fixation to DIN uptake resulted in a positive relationship between delta15N values of Trichormus and the concentration of DIN, because N2 was depleted in 15N relative to DIN. Thus, the differences in delta15N values between Trichormus and Daphnia become smaller with increasing DIN concentration in the Trichormus growth medium and, consequently, it takes longer after die diet switch until isotopic equilibrium in Daphnia is achieved at low DIN than at high DIN concentrations. We used the exponential model of Hobson and Clark (1992) to calculate these turnover rates (https://doi.org/10.2307/1368807). The data reported in Table 2 in the bottom row were from a linear decay model (as was explained in the Table caption) that was used for the highest DIN concentration, because the exponential decay model used for the lower DIN concentrations could not be applied here. However, we agree that due to the lack of differences between the initial and equilibrated isotope values, it makes no sense to calculate an equilibration time at the highest DIN concentration and have therefore deleted the corresponding values from the table. 

2) Throughout paper – Reporting results to two places after the decimal point indicates a level of precision that is not supported. Rounding to one place after the decimal would be more appropriate.

Response: We agree. Has been changed.

3) Lines 61-62. The blanket statement that N2 is depleted in 15N compared to nitrate is not correct. Stable N isotope values in rainwater nitrate are often below zero.

Response: This issue has been also raised by Reviewer 1 with reference to Matiatos et al. 2021, showing that negative delta15N values of nitrate may result from atmospheric nitrate deposition or nitrification in soil (https://doi.org/10.1038/s43247-021-00121-x). We are now citing this paper and included a respective statement on the complexity of isotopic transformations within the nitrogen cycle. 

4) I suggest deleting Figure 3. Since any potential linkage between Daphnia and DIN concentration is via their consumption of Trichormus, Figure 2 should is sufficient. Plotting Daphnia values versus DIN concentrations suggests a direct linkage that does not exist.

Response: That's right, the link between Daphnia and DIN is indirect via the consumption of Trichormus. Nevertheless, we find this relationship interesting, as it illustrates that the effect of DIN availability and thus the switch from nitrogen fixation to DIN uptake is also reflected at the consumer level, which might be of interest also for future field studies. Reviewer 1 suggested to remove the rather meaningless function from Figure 3b (delta13C), and we followed this suggestion.

---

## [Editor Report · Decision Letter 1]

10 Jun 2024

PONE-D-24-09860R1Dissolved nitrogen uptake versus nitrogen fixation: Mode of nitrogen acquisition affects stable isotope signatures of a diazotrophic cyanobacterium and its grazerPLOS ONE

Dear Dr. Martin-Creuzburg,

Thank you for submitting your revised manuscript to PLOS ONE. After careful consideration, we feel that it has merit but does not fully meet PLOS ONE’s publication criteria as it currently stands. Therefore, we invite you to submit a revised version of the manuscript that addresses the points raised during the review process.

 I apologize, but I can't make those simple changes using the editor's platform, at least I don't think so. I appreciate again your efforts to address the reviewer concerns, and look forward to receiving a revised version of the manuscript.

- - - - -

---

## [Author Response · Author response to Decision Letter 1]

11 Jun 2024

Dear Dr. Cooper,

I uploaded a new file "Manuscript_ref revised" with the requested changes to References 22 and 39. 

I apologize for not spotting this while revsing the manuscript.

Best wishes

Dominik Martin-Creuzburg

---

## [Editor Report · Decision Letter 2]

12 Jun 2024

Dissolved nitrogen uptake versus nitrogen fixation: Mode of nitrogen acquisition affects stable isotope signatures of a diazotrophic cyanobacterium and its grazer

PONE-D-24-09860R2

Dear Dr. Martin-Creuzburg,

Thank you for addressing my final concerns with the two references. I'm pleased to inform you that your manuscript has been judged scientifically suitable for publication and will be formally accepted for publication once it meets all outstanding technical requirements.

Within one week, you’ll receive an e-mail detailing any required amendments. When these have been addressed, you’ll receive a formal acceptance letter and your manuscript will be scheduled for publication.

Kind regards,

Lee W Cooper, Ph.D.

Section Editor

PLOS ONE

---

## [Editor Report · Acceptance letter]

24 Jun 2024

PONE-D-24-09860R2 

PLOS ONE

Dear Dr. Martin-Creuzburg, 

I'm pleased to inform you that your manuscript has been deemed suitable for publication in PLOS ONE. Congratulations! Your manuscript is now being handed over to our production team.

Kind regards, 

on behalf of

Dr. Lee W Cooper 

Section Editor

PLOS ONE